# Reciprocal Interactions between Circadian Clocks, Food Intake, and Energy Metabolism

**DOI:** 10.3390/biology12040539

**Published:** 2023-03-31

**Authors:** Emma Grosjean, Valérie Simonneaux, Etienne Challet

**Affiliations:** Institute of Cellular and Integrative Neurosciences, CNRS UPR3212, University of Strasbourg, 67000 Strasbourg, France; emma.grosjean@unistra.fr (E.G.); simonneaux@inci-cnrs.unistra.fr (V.S.)

**Keywords:** circadian rhythm, clock gene, light, feeding time, calorie restriction, high-fat diet, obesity, diabetes, chrononutrition

## Abstract

**Simple Summary:**

Our daily life follows 24-h cycles in various biological functions, such as sleep–wake and feeding/fasting cycles. These rhythms are regulated by endogenous clocks that are synchronized by environmental cues. In mammals, the main circadian clock is located in the suprachiasmatic nuclei at the basis of the brain and can be reset by ambient light, be it natural or artificial. Many secondary clocks, located in the various brain areas and peripheral organs, are regulated by the main circadian clock via the autonomic nervous system, hormonal secretions, and also by meal times. Numerous studies have underlined the importance of circadian rhythmicity for good metabolic health. Moreover, circadian disruption has a negative impact on metabolism, with increased metabolic risks, such as obesity and diabetes. Based on research with animal models and clinical and epidemiological studies in humans, some mechanistic insights are given to explain why circadian disruption, such as exposure to bright light at night or nocturnal meals, can have detrimental effects on energy balance in humans, and how timed nutrition and sleep may counteract or limit these adverse health effects.

**Abstract:**

Like other biological functions, food intake and energy metabolism display daily rhythms controlled by the circadian timing system that comprises a main circadian clock and numerous secondary clocks in the brain and peripheral tissues. Each secondary circadian clock delivers local temporal cues based on intracellular transcriptional and translational feedback loops that are tightly interconnected to intracellular nutrient-sensing pathways. Genetic impairment of molecular clocks and alteration in the rhythmic synchronizing cues, such as ambient light at night or mistimed meals, lead to circadian disruption that, in turn, negatively impacts metabolic health. Not all circadian clocks are sensitive to the same synchronizing signals. The master clock in the suprachiasmatic nuclei of the hypothalamus is mostly synchronized by ambient light and, to a lesser extent, by behavioral cues coupled to arousal and exercise. Secondary clocks are generally phase-shifted by timed metabolic cues associated with feeding, exercise, and changes in temperature. Furthermore, both the master and secondary clocks are modulated by calorie restriction and high-fat feeding. Taking into account the regularity of daily meals, the duration of eating periods, chronotype, and sex, chrononutritional strategies may be useful for improving the robustness of daily rhythmicity and maintaining or even restoring the appropriate energy balance.

## 1. Circadian Rhythmicity

### 1.1. The Circadian Multi-Oscillatory System

The environment in which we live exhibits many variations, such as light exposure, ambient temperature, and food availability, among others. Living organisms adapt to respond to these environmental changes using a set of endogenous clocks located in various parts of the body, which allows a synchronization at the levels of genes, cells, organs and the whole organism. Self-sustained oscillations follow a so-called circadian rhythmicity (i.e., rhythms close to 24 h), which etymologically refers to the Latin terms circa (around) and dies (day). In mammals, there is a main circadian clock, located in the suprachiasmatic nuclei of the hypothalamus (SCNs), which drives the phase of the other secondary circadian clocks dispersed in the body, including various brain areas and peripheral organs. The SCNs are themselves synchronized to 24 h by external cues (time-givers or Zeitgebers in German). The daily variation in ambient light perceived by the retina is the most potent synchronizer of the SCN clock. Besides light (or photic cues), other so-called non-photic cues, including temperature, exercise, and food signals, can also synchronize the circadian clocks. These environmental cues can be predictable (i.e., they are perpetuated every day) or not.

Most behavioral and physiological functions are deeply influenced by circadian rhythms, and conversely, those functions can impact the circadian system. This review will discuss the mutual interactions between energy metabolism and food intake and the circadian system and will highlight how these interactions can be used as chrononutritional strategies to improve or maintain metabolic health [1].

### 1.2. Molecular Machinery

Circadian rhythms are endogenous, innate, and entrainable. They exhibit a period close to 24 h, that is, approximately the duration of an astronomical day. Circadian rhythms are observed in organisms housed under constant lighting conditions (dark–dark or light–light).

As shown in Figure 1, circadian rhythmicity depends on complex molecular machinery with interconnected transcriptional and translational feedback loops involving clock genes and proteins, allowing the generation of self-sustained oscillations. The transcription factors CLOCK and BMAL1 are central to the clock machinery. The CLOCK/BMAL1 complex binds to E-box sequences to promote the expression of the clock genes *Period* (*Per1* and *Per2*) and *Cryptochrome* (*Cry1* and *Cry2*) as well as clock-controlled genes defining circadian outputs. In turn, PERs and CRYs form protein complexes, which, once translocated into the nucleus, interact with the CLOCK/BMAL1 complex to repress its transcriptional activity on the clock and clock-controlled genes. The inactivation of PERs and CRYs by ubiquitination then releases their inhibitory effect on CLOCK/BMAL1 transactivation, thus allowing the restart of a new circadian cycle. Additionally, other transcriptional activators (RORs) and repressors (REV-ERBs) regulate the transcription of *Bmal1* and *Clock* through their RORE promotor sequence. Note that the phosphorylation of clock proteins such as CRY2, PER2, CLOCK, and BMAL1, for instance, by glycogen synthase kinase (GSK) 3β, modulates their stability and triggers their rhythmic degradation. Other ancillary loops exist, such as the additional feedback loop with DEC1 (also called STRA13, SHARP2, or BHLHE40) and DEC2 (SHARP1 or BHLHE41), that repress their own expression via interactions with CLOCK/BMAL1. Additionally, the CLOCK/BMAL1 dimer activates the transcription of albumin D-site binding protein (DBP). DBP activates the transcription of targeted genes via D-box sequences, while E4 promoter-binding protein 4 (E4BP4) suppresses the transcription of the same targeted genes by antiphasic binding to D-boxes [2,3].

In addition to transcriptional and posttranscriptional processes, the molecular clockwork involves a number of posttranslational and epigenetic mechanisms that fine-tune the circadian oscillations and increase their robustness. Notably, large changes in temperature do not impact the speed of the circadian clocks, as is the case for other biochemical mechanisms. This phenomenon, called temperature compensation, prevents the endogenous period of the circadian oscillations from being modified by changes in ambient temperature, thus maintaining accurate internal timing. The molecular mechanisms underlying temperature compensation involve the phosphorylation and degradation of the clock protein PER2 [4].

### 1.3. Synchronization of the Master Clock by Light

An internal clock cycle is most often different from the 24-h day in the environment. Zeitgebers are, therefore, essential to synchronize this endogenous rhythmicity to the external environment. The SCNs are hierarchically the master clock of the other molecular clocks in the brain and peripheral tissues, acting as a conductor that provides a circadian tempo to the secondary clocks of the body. In mammals, light perceived by the retina is the most powerful Zeitgeber of the SCNs. More specifically, light intensity is detected by melanopsin-expressing ganglion cells. These blue-sensing cells transmit light information to the SCNs via the retinohypothalamic tract that releases glutamate and pituitary adenylate cyclase-activating protein (PACAP) in the ventral region of the SCNs. A subsequent increase in intracellular calcium induces the acute transcription of *Per1* and *Per2* genes during the biological night (i.e., when their circadian expression is low) through binding to CRE sequences in their promotors of phosphorylated Ca^2+^/cAMP-response element binding protein (P-CREB). On a daily basis, light signals may differ in timing, duration, intensity, and wavelength, and these parameters influence the way the SCNs will be synchronized [1]. As detailed below (see Section 3.1), metabolic factors and meal times can differentially phase-shift the master clock and/or secondary clocks.

## 2. Effects of Circadian Rhythms on Metabolism

### 2.1. Circadian Rhythmicity in Energy Metabolism

Energy metabolism includes the processes underlying the ingestion of food, the use of nutrients to provide energy, and the storage of reserves to meet energy needs in the absence of energy intake. Metabolic activity is closely governed by circadian clocks so that it is partitioned on a daily basis, in phase with the sleep/wake cycle. Overall, the active phase is concomitant with high metabolic activity associated with physical activity (leading to increased energy expenditure and the depletion of glycogen stores in the active muscles) and with food intake associated with energy supply and the restoration of energy stores (glycogenesis in the liver and muscles and lipogenesis). During the active phase, glucose is the preferred energy substrate for cells. In contrast, the sleeping phase is a period of fasting, concomitant with the utilization of energy stores (glycogenolysis, lipolysis). During the inactive phase, the proportion of lipids in energy consumption increases due to the mobilization of fat stores. Because the daily variations of metabolism are associated with the sleep–wake cycle, the rhythms in metabolic processes (e.g., glycogenesis, lipolysis) are opposite in phase between diurnal and nocturnal species [5]. 

Besides food intake, other factors contribute to the energy balance and daily variations in energy metabolism, including physical activity, resting metabolic rate, and diet-induced thermogenesis. Daily rhythmicity of physical activity, which is controlled by the SCN clock, is one of the most studied circadian rhythms in both rodents and humans. Physical performances, such as muscular strength and exercise capacity, vary in humans according to times of day and types of exercise. Most often, physical performances are higher in the afternoon and evening compared to the morning [6,7,8]. The capacity to dissipate heat during exercise changes in parallel; that is, it is lower in the morning than in the evening. Accordingly, dissipation of exercise-induced thermogenesis can participate in the daily variations of physical performances [9]. Another metabolic parameter that modulates exercise capacity is the fuel substrate (e.g., glucose, triglycerides and/or fatty acids), whose availability depends on both the time of day and nutritional status (i.e., fed or fasted; for review, see [10,11]).

Resting metabolic rate, or resting energy expenditure, defines the number of calories required to maintain physiological processes in an unfed, resting state. In humans under controlled laboratory conditions, resting metabolic rate displays robust circadian variations, with higher values in the late biological day and lower values in the late biological night [12].

Diet-induced thermogenesis (DIT), also known as the thermic effect of food, refers to the heat produced in response to meal ingestion. DIT is due to both obligatory and optional components. The obligatory portion consists of the heat generated by digestion, absorption, and food processing. The facultative part consists of regulated heat production to dissipate food energy. Various gastrointestinal functions such as gastric emptying and intestinal absorption (peptides, lipids, carbohydrates) follow circadian rhythmicity, with maximal efficiency in the early active phase [13]. Furthermore, other hormones that can impact the optional DIT, such as adrenaline or noradrenaline, have higher levels at the beginning of the active phase. As schematized in Figure 2, DIT in humans is considered higher in the morning compared to the evening [14,15,16]. These daily variations in DIT, however, have been recently challenged because they can be explained by circadian changes in the resting metabolic rate [17].

### 2.2. Circadian Rhythmicity in Food Intake

Nutritional homeostasis involves foraging, food intake, and the digestion of nutrients to balance energy intake with energy expenditure. The body needs a certain amount of essential nutrients, such as proteins, carbohydrates, lipids, vitamins, and minerals, to function optimally, while daily metabolic and physical activities decrease the nutrient reserves. Food intake is intended to meet the nutritional requirements for sustaining basic physiological functions, providing energy for physical activity, and restoring energy stores. In addition, food preferences and habits influence food intake [18,19].

Dietary homeostasis is primarily regulated by two brain areas: the mediobasal hypothalamus and the caudal brainstem. During a fasted state, as occurs during the sleep period, the body mobilizes its energy stores. This depletion triggers orexinergic signals, in particular via ghrelin, a hormone released by the mucous membrane of the stomach, which activates sensory neurons in the arcuate nuclei of the hypothalamus (ARC). These sensory neurons contain Agouti-related peptide (AgRP) and neuropeptide Y (NPY), which stimulate the release of orexins and the melanin-concentrating hormone (MCH) from the lateral hypothalamus (LHA). In response to these neuroendocrine signals, the demand and search for food increases. After the ingestion of nutrients during a meal, the B-cells of the pancreas release insulin, while the white adipose tissue releases leptin, which both transmit anorectic signals via the stimulation of pro-opiomelanocortin (POMC)-containing neurons and the inhibition of AgRP- and NPY-expressing neurons in the ARC. Additionally, the hormones glucagon-like peptide 1 (GLP1), released from the stomach, and cholecystokinin, released from the intestine, both exert an anorectic effect. Energy homeostasis is then rebalanced until a new cycle [20].

Like the sleep–wake cycle, the feeding/fasting cycle follows a daily rhythm resulting from a combination of homeostatic needs and temporal control. In humans, daytime feeding is broken into two or three meals, even for temporally isolated individuals [21]. In nocturnal rats, a 24-h fast started at different times of the day does not lead to a quantity of food intake during refeeding that is proportional to the duration of fasting. Instead, refeeding in 24-h fasted rats shows daily rhythmicity, with a peak at the end of their active phase (i.e., at dawn), thus demonstrating the power of circadian control on daily feeding [22]. In humans, the feelings of hunger and satiety display daily rhythms, with a decrease in satiety and an increase in hunger in the evening and, conversely, in the morning [23,24]. A ghrelin peak in the evening and a peak of gastrointestinal peptide YY in the morning may contribute to the daily variations of feelings of hunger and satiety, respectively [25].

Lesions of the SCN clock in rodents abolish the feeding/fasting cycle [26], as they do for the sleep/wake cycle. Feeding rhythm, however, is not driven passively by the daily alternation of active/inactive behaviors. Instead, besides the likely involvement of the SCN clock, feeding rhythm is also controlled by the circadian timing of nuclei in the metabolic hypothalamus [27]. In particular, the circadian clock in the ARC participates in the daily variations of hunger and leptin signals [28]. Furthermore, the circadian clock in the paraventricular nuclei of the hypothalamus (PVN) is involved in the daily rhythmicity of energy expenditure [29].

Besides homeostatic control, hedonicity is also involved in the regulation of metabolism and food intake. The reward circuit in the brain promotes the ingestion of highly palatable foods that can exceed homeostatic metabolic needs [30]. The reward circuit is composed of the ventral tegmental area (VTA), which contains dopaminergic neurons that project to the accumbens nuclei, the amygdala, and other cortical areas. Of interest, the daily variations of hedonic appetite in mice are in antiphase to the homeostatic appetite rhythm and depend on the dopaminergic circadian clock in VTA [31].

### 2.3. Interactions between the Circadian Clockwork and Intracellular Metabolism

As illustrated in Figure 1, circadian clocks rely on intracellular mechanisms that generate temporal cues within the cells. Moreover, multiple, sometimes reciprocal, interactions have been demonstrated between clock factors and the regulators of intracellular metabolism. Some examples illustrating these functional links are described below.

First of all, some components of the molecular clockwork may have metabolic functions. RORs and REV-ERBs are circadian transcription factors that also regulate the expression of genes regulating lipid metabolism in peripheral tissues [32,33,34]. The circadian transcription factor CLOCK is also a histone acetyltransferase that can acetylate various proteins, including its circadian partner, BMAL1, and the glucocorticoid receptor [35,36]. 

Second, numerous regulators of intracellular metabolism are clock-controlled proteins. Additionally, some of these metabolic regulators can provide feedback to the molecular clockwork to modulate its rhythmicity. Among them are transcription factors activated by fatty acids, such as the peroxisome proliferator-activated receptors (PPARs), which are considered nutrient sensors. Notably, PPAR alpha, a regulator of fatty acid beta-oxidation and inflammation, is a clock-controlled protein whose transcriptional activity is modulated by direct interactions with the clock protein PER2 [37,38]. In turn, PPAR alpha modulates the rhythmic transcription of *Bmal1* [37]. PPAR gamma is a potent regulator of lipid metabolism in the adipose tissue, and its transcriptional activity is repressed by PER2 [39].

Reduction–oxidation (redox) reactions, closely linked to the fed/fasted states of the cells, are also intertwined with the molecular clockwork. The circadian heterodimer CLOCK/BMAL1 regulates the rhythmic transcription of *Nampt*, coding the enzyme nic-otinamide phosphoribosyltransferase that controls the synthesis of nicotinamide adenine dinucleotide (NAD), while NAMPT, in turn, modulates the molecular clockwork [40,41]. Within the master clock, redox cycles participate in the circadian rhythm of the electrical activity of SCN neurons [42].

Sirtuin 1 (SIRT1) is a histone deacetylase dependent on the NAD+ that influences the phase of circadian clocks and increases their amplitude, including in the SCNs [40,43,44]. AMP-activated protein kinase (AMPK), another metabolic sensor whose activity is enhanced by the low availability of ATP, such as fasting, interacts with PER2 and CRY1 degradation and, thus, modulates the phase of the cellular clocks [45,46]. Another intracellular metabolic sensor is the mammalian/mechanistic target of rapamycin (mTOR), which is activated by feeding signals and interacts with PER1, thus changing the speed of the circadian clocks [47,48].

The aforementioned, non-exhaustive molecular mechanisms provide additional regulatory feedback loops within the circadian clocks and define very tight connections between circadian rhythmicity and intracellular metabolism. Such molecular interactions lead to many possible pathways by which signals related to food intake, energy availability, exercise, and thermoregulation can impact circadian clocks.

### 2.4. Circadian Disruption Affects Metabolism

Circadian alterations can be caused by several factors, including genetic mutations or inappropriate environmental cues, both affecting the clockwork or its synchronizing properties. As illustrated below, whatever the primary cause, circadian disruption has a negative impact on metabolic health.

#### 2.4.1. Genetic Causes

Mutations or global deletions of a clock gene in rodents are associated with metabolic abnormalities, such as diabetes, obesity, hypertension, and atherosclerosis, that together contribute to cardiometabolic syndrome [49,50,51,52]. The etiology of these metabolic disorders is not always easy to define because they may depend on systemic or local causes or a combination of both. For instance, *Rev-erb alpha* knock-out (KO) mice display mild hyperglycemia and increased adiposity. While their fatty acid utilization is increased during the daytime, the fat phenotype in *Rev-erb alpha* KO mice is mainly attributed to increased lipogenesis from dietary carbohydrates at night [51]. Furthermore, tissue-specific clock disruption in a single peripheral organ, such as the liver or the pancreas, is sufficient to trigger metabolic disorders [52,53,54]. In humans, it is worth noting that various nucleotide polymorphisms in clock genes, such as *Bmal1, Clock, Cry1-2,* and *Rev-erb alpha*, have also been associated with metabolic disorders [55,56,57,58]. A meta-analysis confirms an association between clock gene polymorphism and cardiometabolic syndrome [59].

#### 2.4.2. Mistimed Light Exposure

Compared to relatively rare genetic causes, the environmental origin of circadian disruption cannot be neglected nowadays. Indeed, a number of societal activities occur 24 h a day and 7 days a week, requiring humans to work in shifted conditions often associated with exposure to light at night and to meals taken at night, thus during the period of sleep. Ambient light is the most powerful synchronizer of the master clock, and exposure to bright light at night can phase-shift all the secondary clocks and rhythms that the SCN clock controls. In the same line, nocturnal meals can phase-shift the peripheral clocks. Meta-analyses of human studies report significant associations between shift work and the risk of developing cardiometabolic syndrome [60,61,62]. In animals, chronic jetlag caused by repeated shifts in the light–dark cycle, thus mimicking shift work conditions, affects metabolic health. Chronic jetlag in nocturnal rats favors body mass gain and impairs insulin regulation [63,64]. In diurnal rodents, chronic jetlag not only impairs glucose tolerance but also accelerates cellular aging [65]. Exposure to light pollution at night can also induce health problems. In rats, short-term overnight exposure to light (acute treatment) leads to adverse effects, such as decreased glucose tolerance [66]. More chronic exposure to dim light at night (i.e., for at least 2 weeks) in male rats leads to increased lipid storage in the liver and severely impairs the daily expression of many metabolic genes in the liver and white adipose tissue [67,68]. Furthermore, chronic exposure to dim light at night in female rats flattens the rest–activity rhythm, reduces nocturnal food intake, blunts the estrous cycle, and triggers anhedonia [69]. In humans, a longitudinal study has revealed that long-term exposure to light during the night period is associated with increased atherosclerosis [70]. Finally, a transient circadian desynchronization mimicking jetlag due to air flights across 12-h time zones adversely affected cardiovascular regulation in healthy adults during the following days [71].

#### 2.4.3. Mistimed Sleep/Wake Cycle

Changes in the timing of the sleep/wake cycle and sleep duration can also have detrimental metabolic effects, including inflammatory and diabesogenic consequences [72]. The forced desynchrony protocol in humans kept in laboratory conditions permits disentangled circadian and behavioral effects. Such experimental circadian misalignment induces marked hormonal changes, including the reversed rhythm of plasma cortisol, increased plasma insulin, and reduced plasma leptin [73]. An animal model of human shiftwork, in which rats are forced to be active during the light phase, leads to a loss of plasma glucose rhythm and a reversed rhythm of plasma triglycerides [74]. Notably, social jetlag, defined as a slight change in the sleep–wake cycle during weekends and free days compared to working days, has been positively correlated with an increase in body mass index (BMI) [75,76].

#### 2.4.4. Mistimed Eating

Besides or in combination with the detrimental metabolic effects of light at night, mistimed meals (i.e., those taken during the resting phase) have negative influences on metabolic health.

In rodents, imposed access to a balanced diet only during the resting phase has desynchronizing and obesogenic effects associated or not with relative hyperphagia [77], [78]. Similar temporal restricted access to an unbalanced (i.e., high-fat) diet leads to more salient obesogenic consequences in nocturnal rodents fed during the daytime compared to those fed with the same diet at night only [79,80,81].

In humans, several parameters of eating behavior can be taken into account when considering mistimed eating, such as the duration and phase of the temporal window of eating. The longer it is, the shorter the subsequent fasting period is. For instance, a US survey reported that during a period of a day, the fasting (sleep) period is reduced to a 6 h period on average, while the eating period spreads over 18 h [82]. On the one hand, a short duration of sleep may, by itself, be detrimental to metabolic health [83]. On the other hand, a late dinner is associated with a short duration of overnight fasting, which, by itself, has a negative metabolic impact. In a Spanish study, late dinners were associated with higher BMI, higher plasma triglycerides, and lower insulin sensitivity compared to subjects having early dinners [84]. Late dinners relative to melatonin onset are significantly associated with a higher percentage of body fat and BMI [85]. Under controlled laboratory conditions, a late isocaloric dinner favors lipid storage and increases hunger feelings [86]. Late evening snacking decreases lipid oxidation during the following overnight fast [87].

Even if a long duration (≤12 h) of daily fasting can be beneficial for metabolic health, breakfast skipping, which increases the duration of overnight fasting, may not be recommended, suggesting a trade-off between appropriate meal timing (i.e., not too late) and duration of overnight fast (i.e., not too short). Breakfast skipping may or may not be associated with increased risks of overweight and diabetes [88,89,90]. Eating jetlag (also called metabolic jetlag), which refers to the possible change in the phase of the temporal window of eating between free days and working days, is positively correlated to BMI, independently of the social jetlag mentioned earlier [82,91]. A cross-sectional study indicates that eating jetlag is mainly due to a phase delay of breakfast on free days.

One way to investigate the role of the daily feeding/fasting cycle in rodents is to expose individuals to ultradian 6-meal schedules, with short food access provided every 3 or 4 h. This protocol shows that the daily timing of metabolic gene expression in peripheral organs is markedly affected without a major phase-shift of peripheral circadian clocks [92,93,94]. 

Together, these data suggest that in humans, keeping timed meals, eating sufficient calories for breakfast and lunch, taking a light early dinner, and having a long overnight fast are beneficial for maintaining metabolic health.

## 3. Effects of Metabolism on Circadian Rhythms

Various components of energy metabolism, including physical activity, changes in body temperature, and meal times, can have resetting properties on circadian clocks. Besides its timing, the quality and quantity of food intake can also modulate the function of circadian clocks but not necessarily their phase (as shown in Figure 3).

### 3.1. Synchronization of Circadian Clocks by Metabolic Cues

#### 3.1.1. Synchronization of the SCN Clock

Besides the ambient light evoked earlier (see Section 1.3), non-photic factors can also shift or entrain the SCN clock, especially in the absence of or under a weak light–dark cycle. These non-photic Zeitgebers of the SCNs include exercise and behavioral arousal that can phase-shift the SCNs during the resting phase and modulate their synchronization to light. The neurochemical pathways that convey exercise-related cues to the SCNs comprise the serotonergic projections from the raphe nuclei, NPY-containing projections that originate in the intergeniculate leaflets of the thalamus (IGL), and cholinergic inputs coming from the basal forebrain. Orexigenic neurons of LHA also provide arousal-promoting cues via projections to the SCN neighborhood. In the SCN clockwork of nocturnal rodents, phase-shifts associated with behavioral activation imply a transient down-regulation of the clock genes *Per1* and *Per2* during the biological daytime (i.e., when their circadian expression is high) [95,96].

Contrary to peripheral clocks (see Section 3.1.2), the daily variations in ambient temperature are weak Zeitgebers for the SCNs in most mammalian species. Such resistance to temperature entrainment has been attributed to the strong intercellular coupling of the SCN [97]. Ambient temperature cycles, however, can entrain the master clock in at least two mammals adapted to desert conditions, namely, the camel and the desert goat [98,99]. The molecular mechanisms explaining the SCN sensitivity to temperature cycles in these species are not yet fully understood.

Timed feeding is not a potent time-giver for the SCN clock if the animals or human subjects are exposed (i.e., synchronized) to a light–dark cycle. In these conditions, the master clock remains synchronized to light [100,101]. However, this relative insensitivity to feeding synchronization does not mean that the SCNs receive no nutritional cues that may affect their daily functioning (see Section 3.2 for details). Furthermore, in the absence of photic synchronization (i.e., constant darkness or constant lighting), the SCNs can be entrained by daily restricted feeding in several strains of mice [102,103,104].

#### 3.1.2. Synchronization of the Peripheral Clocks

The robustness of the peripheral clocks is lower and their self-sustaining oscillations are weaker than those of the master clock [105]. The synchronization of the peripheral clocks by the SCNs, defining an internal synchronization, is thus essential. This is made possible thanks to neuronal, endocrine, and behavioral temporal signals, which may differ depending on the target organ. The circadian clocks of the liver and kidneys, for example, appear to be more sensitive to time-giving signals transmitted by behavioral or blood-borne cues as compared to the circadian clocks in other peripheral tissues such as the skeletal muscles, the heart, and the spleen [106]. Some hormones, such as melatonin and glucocorticoids, whose rhythms are closely synchronized by the SCN clock, have a powerful time-giving effect on secondary clocks expressing their respective receptors [107].

One of the major questions regarding this internal synchronization between SCNs and peripheral clocks lies in the temporal switch between nocturnal and diurnal species. Several studies indicate that functional differences may occur downstream of the main clock, which is always most active during the daytime regardless of the timing of the active phase. Therefore, the same signals coming from the SCNs may be interpreted differently by the target structures between diurnal and nocturnal animals. For instance, the daytime release of SCN vasopressin in paraventricular hypothalamic nuclei (PVN) has stimulatory and inhibitory effects on the release of adrenal corticosterone in diurnal and nocturnal rodents, respectively [108]. 

Although peripheral clocks do not receive direct photic cues, they can maintain daily rhythmicity under a light–dark cycle, even if the master clock is deficient. This temporal organization may be maintained by the PVNs that receive photic signals and would then transmit timing signals via the sympathetic system [109]. The same pathways may convey the stimulating and inhibitory effects of ambient light on adrenal corticosterone synthesis and pineal melatonin synthesis, respectively [110,111].

Time-restricted feeding is a potent synchronizer of most peripheral clocks, including the liver and even the skin [100,101,112]. This property has been evidenced in rodents by limiting food access to a few hours per day, notably during the inactive period, to better highlight feeding-induced phase-shifts in clock gene expression. However, not all peripheral clocks respond in the same way to timed food intake. While the liver clock is tightly synchronized by the feeding period, independently of the photoperiodic conditions, the pituitary clock integrates both feeding and photic synchronizing signals, thus leading to a reduced phase-shift in mice with daytime-restricted feeding [113]. In the same line, daytime-restricted feeding in rats has strong synchronizing effects on the liver clock but weak effects on the brown adipose tissue clock. Furthermore, daytime-restricted feeding does not shift the daily expression of clock genes in the skeletal muscle clock, which is nevertheless markedly disturbed [93]. Investigations in humans kept under controlled laboratory conditions show that timed meals can phase-shift peripheral rhythms, such as body temperature rhythm, plasma glucose rhythm, and adipose clock gene expression, while not affecting SCN phase markers such as the nocturnal rhythm of plasma melatonin [114,115]. 

Food intake induces numerous metabolic and hormonal changes in the blood that may contribute to the synchronizing effects of time-restricted feeding. Metabolites, such as glucose, hormones, such as pancreatic insulin, and intestinal gluco-incretins, such as oxyntomodulin and glucagon-like peptide 1, released after food ingestion, may act as post-feeding timers for circadian clocks in peripheral organs. Accordingly, increased glucose availability modulates the circadian clockwork in vitro [116,117]. In the same line, insulin, oxyntomodulin, and glucagon-like peptide 1 can induce phase-shifts of circadian clocks in the liver, kidney, and fibroblasts [118,119,120,121]. On the other hand, metabolites and hormones secreted during the fasting state and sleep may act as pre-feeding timers for peripheral clocks. For instance, increased levels of fatty acids and metabolic transcription factors activated by them, such as PPAR alpha, and increased levels of plasma glucagon modulate the rhythmicity of peripheral circadian clocks [122,123].

Timed exercise is an efficient synchronizer for the peripheral clocks of rodents, notably in the skeletal muscle [124]. In humans, resistance exercise modifies clock gene expression in skeletal muscles. The entraining pathways of exercise on peripheral rhythmicity are not fully identified. Acute expression of *Dec1* and *E4BP4* is among the transcriptional effects induced by exercise in the peripheral circadian clocks [125,126].

Changes in internal temperature, such as hyperthermia and hypothermia, can phase-shift peripheral clocks [97]. Furthermore, artificial temperature cycles entrain peripheral rhythmicity both in vivo and in vitro [127,128]. The intracellular pathways mediating temperature entrainment involve heat shock responses. In particular, the inhibition and induction of heat shock factor 1 (HSF1) by cool and warm pulses, respectively, play a crucial role in shifting the peripheral clocks in response to temperature changes [97,129].

### 3.2. Metabolic Disturbances Affect Circadian Rhythmicity

#### 3.2.1. Calorie Restriction and Negative Energy Balance

Even if the SCN clock is not shifted by time-restricted feeding in animals under a light–dark cycle, it receives metabolic information from the periphery. In response to negative energy balance, some mammals, such as Siberian hamsters, become less active and sometimes torpid. Nocturnal rodents, such as rats and mice, have an opposite strategy because they become hyperactive and partially diurnal, that is, partially active during their usual resting period. This behavioral switch is associated with altered clockwork within the SCNs [130]. Furthermore, the photic regulation of the master clock is modulated by negative energy status. During calorie restriction, light-induced phase-advances in mice are increased in amplitude and more widespread during the daytime [131]. When glucose availability is reduced, light-induced phase-delays in mice are decreased [132]. Additionally, an ultradian 6-meal schedule, associated or not with calorie restriction, affects peripheral clocks as well as the clockwork in the SCNs. These findings indicate that the SCNs receive not only metabolic cues but also feedback signals from the daily feeding/fasting cycle [92].

Among peripheral metabolic signals that can directly reach the SCNs is glucose. Accordingly, SCN neurons harbor glucose sensors through ATP-sensitive K^+^ channels [133]. The availability of fatty acids and the PPAR pathway may also modulate the SCN function [134]. Hormonal cues, such as ghrelin, insulin, and leptin, are considered to provide more indirect signals via the mediobasal hypothalamus [119,135,136].

Calorie restriction, which is known to lengthen lifespan, has strong effects on rhythmic gene expression in the liver and delays age-related changes in hepatic gene expression. Importantly, the positive effects of calorie restriction on longevity in mice are improved if the diet is provided at the beginning of the active phase rather than at the beginning of the resting phase [137].

#### 3.2.2. Obesity and Positive Energy Balance

Whether of genetic or nutritional origin, obesity causes numerous circadian disturbances. In a genetic model of obese mice with a mutation in the *Ob* gene coding leptin, alterations in peripheral molecular clocks are observed even before the first metabolic disturbances. *Ob/ob* mice display an opposite daily rhythm in blood glucose without concomitant shifts in food intake and blood lipids [138]. Regarding the SCN function, *ob/ob* mice show increased light-induced phase-delays, an effect that can be normalized with leptin treatment [136].

Obesity due to high-fat feeding leads to changes in gene expression in peripheral clocks [139]. In mice fed with a high-fat diet, the SCN clock runs faster and its synchronization to light is slowed down [139,140]. Additionally, their spontaneous increase in food intake during the resting phase occurs within the first few days of high-fat feeding, that is, well before the onset of body mass gain [139]. 

In obese humans, less obvious circadian disturbances have been found. For instance, obese women display a more flattened and fragmented rhythm of wrist temperature compared to normal-weight women [141]. The amplitude of the nocturnal secretion of plasma melatonin is increased in obese and diabetic subjects compared to lean control participants [142]. By contrast, the nocturnal secretion of melatonin was found to be lower in women with metabolic syndrome compared to age-matched women without metabolic syndrome [143]. Regarding the temporal distribution of calorie intake, overweight subjects usually consume more of their daily calories for dinner compared to lean controls [144]. As mentioned earlier, no clear conclusion has been reached regarding breakfast skipping in obese people [88,89,90].

#### 3.2.3. Diabetes

*Db/db* mice are obese and heavily diabetic due to a mutation in the leptin receptor. Peripheral oscillations are altered in *db/db* mice [145], while their free-running period, reflecting the speed of the SCN clock, is lengthened compared to *db/+* control littermates [146]. 

In patients with type 2 diabetes, fasting blood glucose is no longer rhythmic as it is in healthy subjects, while the amplitude of their body temperature rhythm and heart rate is diminished [147]. In contrast, neither the phase of the nocturnal secretion of plasma melatonin nor its amplitude differs between weight-matched diabetic subjects and healthy participants [142]. In patients with obesity and diabetes, however, daily rhythms in clock gene and metabolic gene expression in white adipose tissue are dampened compared to those in healthy lean participants [148].

## 4. Chrononutrition and Metabolic Health

Chronotherapy uses the specificity of daily rhythms in biological parameters to improve the treatment of some pathologies [149]. Therefore, depending on the targeted biological parameter, a specific time slot for the therapeutic treatment can be more appropriate in order to obtain better effectiveness and/or lower side effects [150,151]. Similarly, the chronobiological organization of food intake can be particularly beneficial for maintaining good metabolic health.

### 4.1. Time Slot and Regularity

Snacking, i.e., eating outside of main meals, may lead to metabolic complications, including weight gain, especially in the case of poor dietary snacks, although this point is still a subject of debate among nutritionists [152,153]. In addition, snacks are frequently consumed at any time of the day, and this lack of regularity may be one of the causes of weight gain [82]. A regular schedule of meals is, therefore, essential as it allows for better management of energy balance. Eating meals at fixed times from dawn to dusk (and not continuously throughout the day) therefore improves circadian rhythmicity and sets in motion a positive gear for good metabolic health. 

Glucose tolerance and insulin sensitivity follow a circadian rhythm whose peaks occur in the morning. Accordingly, improved glucose tolerance and increased insulin sensitivity in the morning allow for better regulation of carbohydrate-rich breakfasts and lunches [151]. The regularity and timing of food intake are essential not only to maintain a balanced metabolism but also to reinforce its circadian rhythmicity. In healthy subjects, having breakfast, eating the most important meal at the beginning of the day (breakfast or lunch), and avoiding dinner energy intake close to bedtime or late in the evening are preventive actions against body mass gain [154,155].

### 4.2. Shortening of the Daily Period of Eating

A pioneer study on that topic has been performed in Zucker rats, a genetic model of obesity, insulin resistance, and hypertension due to a fa mutation in the leptin receptor. When fed ad libitum with a standard chow diet, Zucker rats are binge eaters during their resting phase. Limiting access to the same balanced diet during the nighttime, which corresponds to the active period in nocturnal rats, results in a decrease in body mass gain without affecting the daily energy intake compared to Zucker rats fed ad libitum [156]. This highlights the appropriateness of a temporal window of feeding restricted to the active period for body weight management. More recent studies using unbalanced diets in mice have also demonstrated that high-fat feeding or high-fat/high-sugar feeding limited to the nighttime or shorter periods at night limits increased adiposity and hepatic steatosis compared to mice fed ad libitum with the same diets [79,157,158].

A number of studies on patients with metabolic disorders, including overweight and obesity, have demonstrated the effectiveness of a diet based on a reduced eating window during the active period. For example, shortening the eating window from 14 h to 10–11 h during the daytime in overweight patients for 4 months led to a decrease in BMI and also an improvement in sleep [82]. On the same principle, obese patients subjected to diets with a timed fasting period of 14 h (including the sleep phase) over 3 months showed a decrease in BMI and a reduction in blood pressure [159]. 

Regarding prediabetic patients, a daily period of eating reduced to 6 h before 3 pm for 5 weeks improved insulin sensitivity and blood pressure, but with no change in BMI [160].

Together, the above examples confirm the benefits of limited access to food during the active period, not only on the regulation of metabolism but also on the general rhythmicity of the body. Accordingly, limiting the eating period to 12 h or even less during the daytime, thus leading to a fasting period (including sleep) of at least 12 h, helps humans to recover or maintain good metabolic health [161,162,163].

### 4.3. Chronotypes

The chronotype is an individual manifestation of circadian rhythmicity that defines the internal preference of a given person for more morning or more evening activities, in particular at bedtime and wake-up time. Similarly, the rhythms of other biological parameters, such as circulating hormones, can occur with phase differences between early (i.e., “larks”) and late chronotypes (i.e., “owls”). The chronotype of a given individual is, therefore, an important biological parameter to take into account in order to manage his/her metabolism in the best way. Various studies have shown that late eaters (i.e., mostly evening chronotypes) are more likely to gain weight and have more difficulty in losing weight, thus increasing their risk of obesity [164]. Of note, significant associations were detected between genetic variants of two clock genes, *Per1* and *Clock,* and late chronotypes and obesity risk [165]. Conversely, morning chronotypes, who consume more calories in the morning, have a significantly reduced risk of developing obesity [166].

### 4.4. Sex Differences

Despite the various chrononutrition strategies to best manage one’s energy balance for keeping a healthy state, sex differences are often overlooked. The differential responses between females and males should be taken into account in a chrononutritional approach in order to best meet the metabolic needs of each sex [167]. Before puberty, boys and girls have a similar metabolism, but starting at puberty, the sex hormones may have differential metabolic effects. Thus, female hormones, especially estrogen, which governs the endogenous menstrual cycle, have an impact on the general metabolism [168]. The menstrual cycle, with an average duration of 28 days, is divided into the follicular phase and the luteal phase (ending with menstruation), separated by ovulation. The pituitary gonadotropins (follicle stimulating hormone (FSH), luteinizing hormone (LH)) and the ovarian hormones (estradiol and progesterone) exhibit large variations during the menstrual cycles. At the beginning of the follicular phase, circulating FSH and estrogen increase gradually. At the end of the follicular phase, the high level of estrogen promotes a large and transitory increase in LH, which triggers ovulation. During the luteal phase, the secretion of progesterone increases while the circulating estrogen and gonadotropins return to baseline levels. If pregnancy has not occurred, the cycle ends with menstruation. Energy metabolism is greatly modulated by this rhythmic female reproductive physiology. Women show higher metabolic rates at the end of the luteal phase than at the end of the follicular phase [169]. Moreover, protein oxidation at rest is greater in women during the luteal phase [170]. These different studies, therefore, indicate an increased metabolism during the luteal phase, which is synonymous with high caloric expenditure, often accompanied by greater nutritional demand. As a result, specific nutritional strategies are to be adapted according to the phase of the menstrual cycle. Furthermore, as estrogen has been described as a regulator of fat mass, glucose homeostasis, and sleep during the (pre)menopausal period, the lack of estrogen favors increased adiposity and cardiometabolic risks [171,172].

In addition, there are sex differences in the regulation of circadian rhythms. In rodents, for instance, there is a sexual dimorphism in the rhythmic gene expression of the liver and its metabolism [173,174]. Sex-dependent activity in the liver is lost in male mice KO for *Cry1* and *Cry2,* displaying a feminized liver phenotype [173]. In humans, the circadian period of the master clock is slightly shorter in women (24.1 h) than in men (24.2 h), and more women have circadian periods shorter than 24.0 h [175]. Furthermore, circadian misalignment does not affect respiratory quotient and metabolic hormones in the same way between men and women. Among others, during circadian desynchronization, the decrease of the respiratory quotient, indicative of a reduced rate of carbohydrate oxidation in favor of an increased rate of lipid utilization, is detected in women only. Furthermore, circadian misalignment down- and up-regulates the rhythmic secretion of plasma leptin in women and men, respectively [176]. 

It is, therefore, necessary to take into account sex in chrononutritional strategies. Notably, since the metabolism and energy needs of women change according to their menstrual status and their reproductive age, the inappropriate timing of intake may not have the expected beneficial effects and could even have adverse impacts on women’s health and metabolism.

## 5. Conclusions

In recent years, many studies have taken advantage of the advanced knowledge acquired about circadian rhythms and metabolism to address the nutritional challenges of a society with growing metabolic disorders. Chrononutrition is one direct application of the mutual interactions between circadian rhythms and metabolism. Although the direct impact of circadian rhythms on metabolism is a notion that is becoming more and more accepted, the reciprocal link should also be considered. As illustrated throughout this review, metabolism, food intake, and physical activity regulate and (de)synchronize circadian rhythms, eventually leading to a vicious circle between circadian disruption and metabolic disturbances. A better understanding of these interactions is a societal issue, and it will be useful to develop new chronotherapeutic dietary approaches for better general health.

## Figures and Tables

**Figure 1 biology-12-00539-f001:**
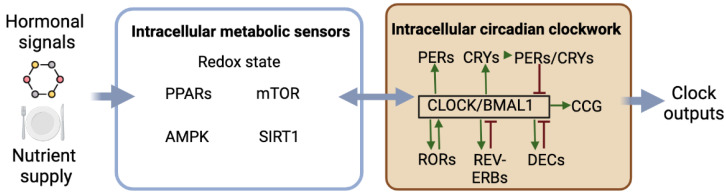
Schematic view of the intracellular machinery of circadian clocks and their interactions with cellular metabolic factors. Nutrient supply, hormonal signals, and physical activity directly influence intracellular metabolic sensors. These factors interact with different genes and proteins of the circadian clock mechanism reciprocally. The clock machinery is composed of different autoregulatory loops of activation and inhibition, providing feedback cues and clock outputs. AMPK, AMP-activated protein kinase; BMAL, brain and muscle aryl-hydrocarbon receptor nuclear translocator-like protein; CCG, clock-controlled genes; CLOCK, circadian locomotor output cycles kaput; CRY, cryptochrome; DEC, differentially expressed in chondrocytes protein; mTOR, mammalian/mechanistic target of rapamycin; PER, period; PPAR, peroxisome proliferator-activated receptor; REV-ERB, reverse viral erythroblastosis oncogene product; ROR, retinoic acid receptor-related orphan receptor; SIRT, sirtuin. Created with https://www.biorender.com/ (accessed on 10 February 2023).

**Figure 2 biology-12-00539-f002:**
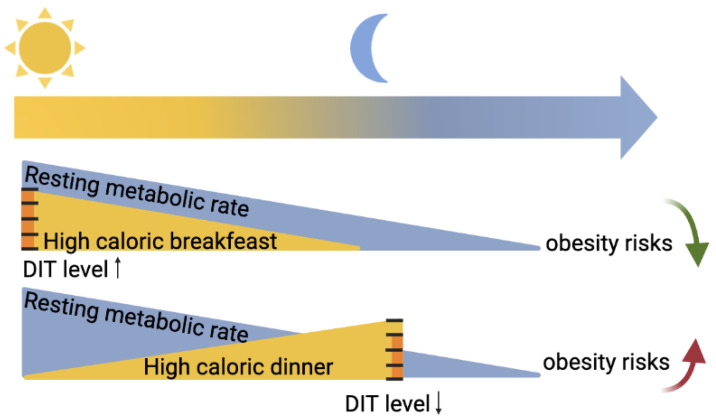
Changes in resting metabolic rate and diet-induced thermogenesis (DIT) over the course of a day. The resting metabolic rate in humans follows a daily rhythm, with higher values during the day than during the night. It is advisable to favor caloric meals when the resting metabolic rate is high, i.e., in the morning, and minimize it when it is lower, i.e., before bedtime. For the same caloric intake, the DIT level is higher in the morning than in the evening. Depending on the characteristics of the DIT, eating in the morning (high DIT) would be more advantageous for weight loss and preventing obesity than eating rich meals in the evening (lower DIT). Created with https://www.biorender.com/ (accessed on 10 February 2023).

**Figure 3 biology-12-00539-f003:**
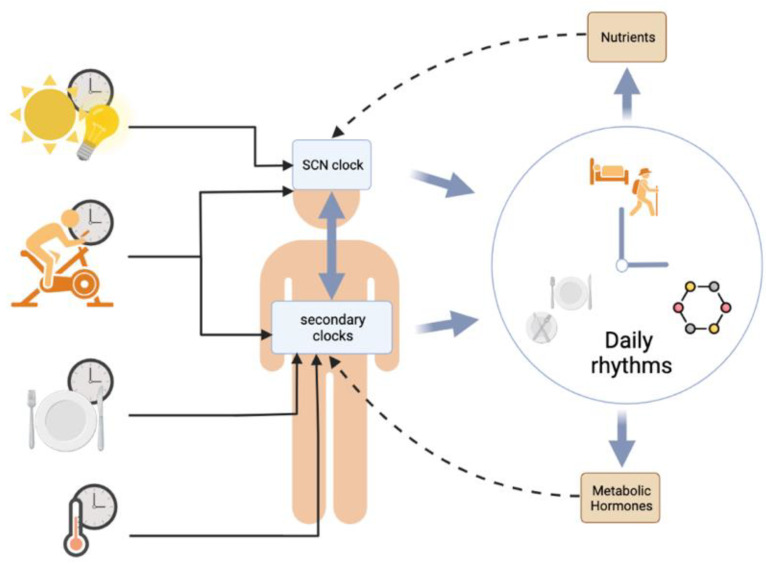
Schematic representation of the reciprocal connections between food intake/metabolic energy and circadian clocks. The main circadian clock, located in the suprachiasmatic nuclei of the hypothalamus (SCN), and the peripheral clocks (brain and peripheral organs) are synchronized by the daily timing of ambient light (SCN), physical activity (SCN and peripheral clocks), meals, and temperature (peripheral clocks). The internal synchronization between these different clocks takes place, producing different daily rhythms such as the sleep–wake cycle, the feeding–fasting cycle, and hormonal rhythms. These rhythms also participate in the regulation of the different clocks, notably through metabolic hormones and nutrient requirements. Created with https://www.biorender.com/ (accessed on 10 February 2023).

## Data Availability

No research data was specifically analyzed to write this review article.

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
