# Peer review of "Reciprocal Interactions between Circadian Clocks, Food Intake, and Energy Metabolism"

_biology, 2023, doi:10.3390/biology12040539_

Round 1

Reviewer 1 Report

Dear Authors,

The theme of the circadian system, its synchronization on the cell level, and the central and peripheral connection are logically presented. It would be of great value to those interested in circadian rhythms and metabolism. It was my pleasure to read your manuscript and I have only minor comments:

1.         I did not find the connection between citations 153 and 154 to the text in the manuscript (lines 583 – 585). I recommend excluding this citation or rewriting the text.

2.         Perhaps the above-mentioned articles should enrich part 4.2.4. The articles are dealing with artificial light at night and metabolism in laboratory rodents:

Rumanova, V. S., Okuliarova, M., Foppen, E., Kalsbeek, A., & Zeman, M. (2022). Exposure to dim light at night alters the daily rhythms of glucose and lipid metabolism in rats. Frontiers in Physiology, 1686.

Okuliarova, M., Rumanova, V. S., Stebelova, K., & Zeman, M. (2020). Dim light at night disturbs molecular pathways of lipid metabolism. International journal of molecular sciences21(18), 6919.

Author Response

We thank Referee 1 for having evaluated our manuscript positively.

Point 1. Misplaced references 153 and 154.

We thank Referee 1 for having noted that the references 153 (Corbalan-Tutau et al. 2011, PMID 21721858) and 154 (Corbalan-Tutau et al. 2014, PMID 22705307) were not at all cited at the right place.

We have reworded sentences of paragraph 3.2.2., as follows :

« In obese humans, less obvious circadian disturbances have been found. For instance, obese women display a more flattened and fragmented rhythm of wrist temperature, compared to normal-weight women (Corbalan-Tutau et al. 2011, PMID 21721858). The amplitude of nocturnal secretion of plasma melatonin is increased in obese and diabetic subjects compared to lean control participants [132]. By contrast, nocturnal secretion of melatonin was found lower in women with metabolic syndrome, compared to age-matched women without metabolic syndrome (Corbalan-Tutau et al. 2014, PMID 22705307). »

Point 2. Additional references on the deleterious effects of light at night.

According to the suggestion of referee 1, we cited the 2 new references proposed (plus another one) on the effects of light at night in section 2.4.2., as follows :

« Exposure to light pollution at night can also induce health problems. In rats, short-term overnight exposure to light (acute treatment) leads to adverse effects such as decreased glucose tolerance [61]. More chronic exposure to dim light at night (i.e. during at least 2 weeks) in male rats leads to increased lipid storage in the liver, and severly impairs the daily expression of many metabolic genes in the liver and white adipose tissue (Okuliarova et al. 2020, PMID 32967195 ; Rumanova et al. 2022, PMID 36105299). Furthermore, chronic exposure to dim light at night in female rats flattens rest-activity rhythm, reduces nocturnal food intake, blunts the estrous cycle and triggers anhedonia (Gutierrez-Perez et al. 2023 PMID 36650949). In humans, a longitudinal study revealed that long-term exposure to light during the night period is associated with increased atherosclerosis [60]. »

Reviewer 2 Report

The authors provide an overview of the interactions between circadian clocks, food intake and energy metabolism. The review is well done and easily readable despite the complexity of the topic. 

Whereas the part on food intake is comprehensive, I believe that the part on energy metabolism would benefit of some more details on exercise (e.g. interaction between timing and type of exercise and circadian clock). 

Author Response

We thank Referee 2 for having evaluated our manuscript positively.

Point 1. Missing sentences on the effect of time of day on exercise capacity.

We agree that the interactions between timing, type of exercise and circadian clock were missing in the introductory section on energy metabolism. We thank Referee 2 for raising this important point.

To fill this gap, we added new sentences in paragraph 2.1., as follows:

« Physical performances, like muscular strength and exercise capacity, vary in humans according to times of day and types of exercise. Most often, physical performances are higher in the afternoon and evening compared to the morning (Wyse et al. 1994, PMID 8000814; Moussay et al. 2002, PMID 12511031 ; Souissi et al. 2004, PMID 14750007). The capacity to dissipate heat during exercise changes in parallel, that is, it is lower in the morning than in the evening. Accordingly, dissipation of exercise-induced thermogenesis can participate to the daily variations of physical performances (Waterhouse et al. 2005 PMID 16021839). Another metabolic parameter that modulates exercise capacity is the fuel substrate (e.g. glucose, triglycerides and/or fatty acids) whose availability depends on both times of day and nutritional status (i.e. fed or fasted ; for review, see Gabriel and Zierath 2019, PMID 30655625; Aoyama and Shibata 2020, PMID 32181258). »